# Quantum Monte Carlo simulations in the restricted Hilbert space of Rydberg atom arrays

Pranay Patil[1*]

**1** Max-Planck Institute for the Physics of Complex Systems, Dresden, Germany
* patil@pks.mpg.de

September 6, 2024

## Abstract

Rydberg atom arrays have emerged as a powerful platform to simulate a number of exotic quantum ground states and phase transitions. To verify these capabilities numerically, we develop a versatile quantum Monte Carlo sampling technique which operates in the reduced Hilbert space generated by enforcing the constraint of a Rydberg blockade. We use the framework of stochastic series expansion and show that in the restricted space, the configuration space of operator strings can be understood as a hard rod gas in $d+1$ dimensions. We use this mapping to develop cluster algorithms which can be visualized as various non-local movements of rods. We study the efficiency of each of our updates individually and collectively. To elucidate the utility of the algorithm, we show that it can efficiently generate the phase diagram of a Rydberg atom array, to temperatures much smaller than all energy scales involved, on a Kagomé link lattice. This is of broad interest as the presence of a $Z_2$ spin liquid has been hypothesized recently [1].

# 1 Introduction

Recent advances in ultra-cold atom arrays in optical traps have generated renewed interest in understanding the possible condensed matter systems which can be realized in such systems. One of the most prominent of these is the Rydberg blockade mechanism, which is engineered by exciting atoms with a single occupancy in the outer $s$-shell to high principal quantum numbers [2–5]. The large electron cloud of such a state generates a strong Coulomb repulsion, which prevents the excitation of atoms in close vicinity. This can be modeled theoretically as an exclusion process, and has led to the so-called $PXP$ model. This strongly interacting model of spins has a variety of interesting features in both its static and dynamic behavior, and has been found to be of interest for frustrated magnetism, quantum many-body scar states and quantum computing [6–10].

Within the context of magnetism, it has been found that the most interesting consequences of this exclusion principle are seen in two- or three-dimensional systems. For the strongly correlated regimes of interest for realizing non-trivial magnetic order, exact analytic treatments are not feasible, and one of the most powerful numerical techniques to simulate such systems is quantum Monte Carlo (QMC). Although the models for Rydberg atoms are similar to transverse field Ising (TFI) models, and thus lack the infamous sign problem [11] which makes QMC unscalable, the phase space structure allowed by the Rydberg blockade is highly correlated and notoriously hard to sample efficiently. There have been systematic developments in the past [12, 13] to sample TFI (including the highly frustrated case [14–17]) and related models such as the quantum dimer model within the framework of stochastic series expansion (SSE-QMC). Of particular relevance to Rydberg atom arrays emulating frustrated magnets are the motif-marking methods [18], the sweeping cluster algorithm [19, 20], local in space cluster updates [17, 18], and directed loop updates [21, 22]. Although these algorithms have been implemented for Rydberg atom arrays [20, 23, 24], the parameter ranges which have been investigated have been relatively modest due to inefficiencies in sampling, which are especially acute at the low temperatures required to see genuine quantum many-body physics.

Here we develop a formalism for SSE-QMC directly in the restricted space generated by the Rydberg blockade by realizing a mapping between the operator string sampled by SSE-QMC and a hard rod model. Versions of the latter have already been studied from the perspective of soft matter physics, where they have been often used to model nematic crystals [25, 26]. Due to the mapping from a quantum model, the hard rod model generated in our case is fundamentally anisotropic, and requires new updates which we describe here. Updates in a similar rod language were also used for a transverse field Ising glass in the context of continuous time QMC [27].

We begin by describing the Hamiltonian and operators relevant for SSE-QMC, and present the $d + 1$-dimensional hard rod model realized in space-time. This is followed by a description of the standard diagonal update, and a detailed discussion of three cluster updates in the rod language which allow us to move efficiently within the phase space of allowed rod configurations. To enhance the efficiency of the cluster updates described, we implement an optimized form of parallel tempering, and provide detailed results on the efficiency of various updates in different regions of parameter space. We follow this up by

showing an application of our algorithm to the case of a Kagomé link lattice, which has recently been proposed to realize a $Z_2$ quantum spin liquid, and study the dependence of relevant order parameters on temperature and the detuning parameter. Our preliminary results find no presence of spin liquid behavior down to temperatures which are 5% of the smallest energy scale in the system. We conclude by emphasizing that our algorithm is lattice independent and by elucidating other cases of interest where it can be applied.

## 2 Model and formulation of SSEQMC

The most common model [1,23] for Rydberg atom arrays is given by hard-core bosons with a strong repulsive interaction and a kinetic term which creates/annihilates bosons. We define our system on a 2D lattice with $N$ sites, and do not specify a particular geometry for our analysis (this can be seen as a 2D version of the model studied in Ref. [28]). For ease of comparison we parameterize the Hamiltonian in the same way as Ref. [1], and reproduce it below for completeness :

$$H = \frac{\Omega}{2} \sum_i (b_i + b_i^\dagger) - \delta \sum_i n_i + \frac{1}{2} \sum_{i,j} V(|i-j|) n_i n_j. \tag{1}$$

The Coulomb repulsion due to the Rydberg blockade is encoded in $V(|i-j|)$, which is known to have the functional form $V(r) \approx 1/r^6$ [4]. As this is extremely short range, we make the approximation that for nearest neighbors $V(r) \to \infty$, and $V(r) = 0$ otherwise. This provides us with an effective model to understand the effect of the Rydberg blockade. We have effectively removed all states from our boson occupation basis which have bosons on nearest neighbors. Within this restricted space, the Hamiltonian simply reduces to $H = \frac{\Omega}{2} \sum_i (b_i + b_i^\dagger) - \delta \sum_i n_i$. It is important to note here that even though it is not apparent from the Hamiltonian, the restricted Hilbert space renders the system genuinely interacting. As there is a freedom of one energy scale, we set $\delta = 1$ for all our numerical results, unless otherwise mentioned. In the restricted space, the action of the chemical potential is to maximize the density of bosons, thus leading to fully packed configurations with no neighboring sites simultaneously occupied by bosons. In contrast, the kinetic term acts by creating resonances between the occupied and unoccupied state, and in many cases leads to a phase continuously connected to the trivial $x$-polarized paramagnet one would expect for a TFI model in the limit of large transverse field.

Stochastic series expansion [12,29] proceeds by expanding the exponential in the partition function $Z = Tr[e^{-\beta H}]$ as $\sum_n \frac{(-\beta)^n}{n!} \sum_{s_n} a_{s_n} Tr[s_n]$, where $s_n$ lists all length-$n$ operator strings of the form $Tr[H_{s_n^1} H_{s_n^2} ... H_{s_n^n}]$, and $a_{s_n} = (\Omega/2)^{n_K} (-\delta)^{n_V}$ denotes the weight generated by the coefficients in the Hamiltonian. Here, $n_K$ and $n_V$ denote the number of kinetic $(b_i + b_i^\dagger)$ and potential $(n_i)$ operators in the string $(n_K + n_V = n)$, and the trace is calculated by sampling over the basis states allowed by the Rydberg blockade condition. All terms in this expansion are positive, as the number of kinetic terms must be even to return the state to itself (enforced by the trace), implying that $n_K$ is even and $(-\beta)^{n_K}$ is positive, and each potential term carries a negative sign, which cancels the effect of the negative sign coming from $(-\beta)^{n_V}$. We use the boson occupation basis for computations, and in this basis the actions of these two types of operators can be pictorially represented as

$$H_K = \frac{\bigcirc\bigcirc\bigcirc}{\bigcirc\bullet\bigcirc} + h.c, \quad H_V = \frac{\bigcirc\bullet\bigcirc}{\bigcirc\bullet\bigcirc}. \tag{2}$$

Note that this representation is in 1D, and for a higher dimensional system, the operators will contain all nearest neighbors, thus enforcing the Rydberg blockade. For the Kagomé

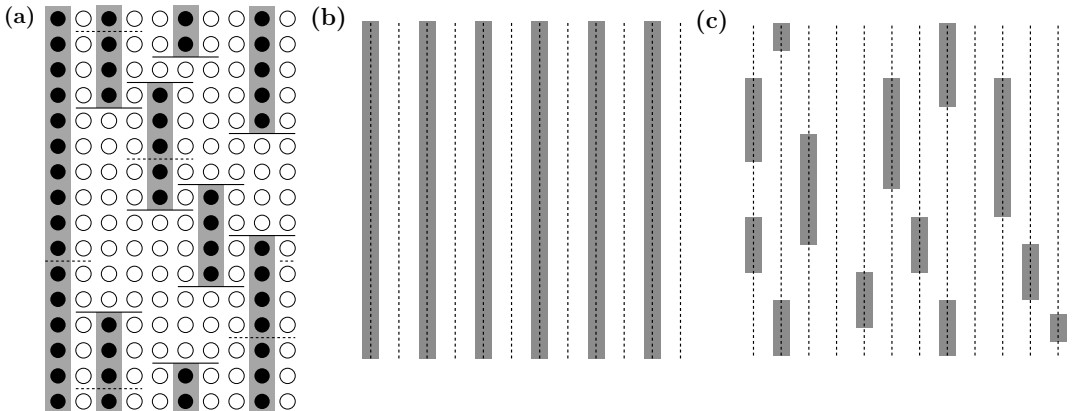

Figure 1: Space-time configurations for a periodic 1D chain with the vertical direction corresponding to imaginary time being also periodic : (a) Operator string using the operator representation defined in Eq. (2), and corresponding rod configuration in gray; (b) Rod configurations dominating in the $\Omega = 0$ limit are made up of fully packed time-spanning rod configurations with no ends; (c) In the $\Omega \gg \delta$ limit, the rods are short and uncorrelated.

link lattice shown in Fig. 2, each site has six nearest neighbors, and thus the operators are defined on 7 spatial sites. In addition to these operators, we also add an energy shift to our Hamiltonian in the form of $-(\Omega/2)I$, as we implement updates which separately exchange $H_V \leftrightarrow I$ and $H_K \leftrightarrow I$ to ensure ergodicity. The global shift does not change the physics and is added in purely to facilitate convenient updates. The negative sign in the coefficient is chosen to remove the sign problem, as already discussed for $H_V$. For ease of computational programming, we also work with a fixed length array to store the operator string, and ensure that we have chosen a length $M$ which is large enough to accommodate relevant fluctuations of the number of operators $n$. The $n$ operators are distributed randomly in $M$ "slices" and the rest of the $M - n$ slices are left empty. This is a standard practice for SSE [30], and is essential for implementing the diagonal update.

It is instructive to also consider the construction of an operator string by evaluating the trace using insertions of complete basis states. This can be understood for a sample operator string $s$ of length three, given by $Tr[H_{s^1} H_{s^2} H_{s^3}]$. Insertions of our basis projector, $\sum_\alpha |\alpha\rangle \langle\alpha|$ between every pair of operators reduces the trace to

$$\sum_{\alpha_1} \sum_{\alpha_2} \sum_{\alpha_3} \langle\alpha_1|H_{s^1}|\alpha_2\rangle \langle\alpha_2|H_{s^2}|\alpha_3\rangle \langle\alpha_3|H_{s^3}|\alpha_1\rangle \,.$$

In most formulations of stochastic series expansion, one uses a projector onto the complete Hilbert space, i.e. the identity operator $I = \sum_i |i\rangle \langle i|$, whereas we restrict the basis states allowed. This restriction implies that not all non-zero matrix elements of the identity $I$, which we have added to the Hamiltonian, are accessed.

Using this notation each operator string can be represented as a boson occupation diagram in one higher dimension. An example of this is shown in Fig. 1 for a one-dimensional system. Note that due to the Rydberg constraint which prevents simultaneous occupation of neighboring sites, regions of the operator string diagram which correspond to occupied sites can be identified as rods (shown with rectangles with dashed borders in Fig. 1) with an exclusion radius of one lattice spacing. Thus configurations of operator strings can be viewed as arrangements of vertical "thick" rods. To get an intuition of the mapping, let us consider the classical ($\delta \gg \Omega$) and quantum ($\Omega \gg \delta$) limits. In the former, the operator string is dominated by $H_V$ and the rods tend to be long as the presence of end

points ($H_K$) is suppressed. In addition, $H_V$ promotes a maximum coverage of space-time by rods, leading to maximally packed configurations. The length of the operator string is controlled by the inverse temperature $\beta$ and in the classical limit, the rods span the entire system in the vertical direction. In contrast, a high density of $H_K$ promotes short rods, and for many lattice geometries, it may be possible to realize a rod paramagnet, where the density of rods is low and their positions are uncorrelated. These limits are shown pictorially in Fig. 1. The sampling of operator strings can now be considered to be a sampling over rod configurations, and we describe in the following section various updates which we use to carry out this sampling, along with benchmarks of efficiency for the novel updates we have introduced. To understand the inherent non-trivial nature of these updates, one can consider a highly dense network of rods which are randomly placed, but respect the Rydberg constraint, and attempt to imagine a controlled modification of rods which will lead to another valid rod configuration. A quick thought experiment is sufficient to convince ourselves that this is not a trivial exercise, especially given that a large penalty in probability must be paid, due to the chemical potential $\delta$, if the rod coverage is modified substantially.

## 3 Monte Carlo updates and efficiency benchmarks

Our simulations are initialized using an empty operator string, and with all the states in the set $\{\alpha_1, ..., \alpha_M\}$ set to the empty (zero boson) state. $M$ is chosen to be $N$ initially, and grown dynamically during the equilibration process to accommodate fluctuations of $n$, until it saturates. We set $M = (4/3)n_{max}$, where $n_{max}$ is the largest value of $n$ seen till the current simulation time. The factor of $4/3$ is chosen to provide sufficient room for any further fluctuations and to allow movement between operators in the diagonal update. A configuration for our simulation is defined by a particular operator string of length $M$ (including empty locations), and a particular choice for the set $\{\alpha_1, ..., \alpha_M\}$, and the probability of this configuration is $\propto \frac{1}{^MC_n} \frac{(\beta^n)}{n!} \delta^{n_V} \left(\frac{\Omega}{2}\right)^{n_K}$. The combinatorial factor here appears as the expansion of $n$ to $M$ creates an unnecessary degeneracy in the operator string configurations over $M$ slices, which correspond to the same string. This must be suppressed to avoid spurious weight factors. The number of equilibrium steps are chosen to be large enough, such that further increments do not change our results, and the updates we use are described below. Although our updates are applicable to general lattices, for ease of understanding, we illustrate most of them in the context of a 1D chain, except for the rod diffusion update, where the 2D case requires important considerations which are absent in the 1D case. However, the measurements of efficiency of various updates are done on the 2D Kagomé link lattice, also known as the ruby lattice. We make this choice to showcase the applicability of our updates to frustrated systems which are expected to be the most difficult to simulate using traditional algorithms.

The ruby lattice (shown in Fig. 2) is generated by considering Rydberg atoms living on the links of a Kagomé lattice [1]. An excited Rydberg atom can be understood as a dimer occupying the link under consideration, and the Rydberg constraint then reduces to a hard-core dimer constraint, which requires that dimers do not share ends. In the classical limit ($\Omega = 0$), the ground state can then be understood to be a fully-packed dimer liquid with paramagnetic correlations [31, 32]. This model is expected to host a $Z_2$ spin liquid at zero temperature at intermediate $\Omega/\delta$. We use a unit cell of six sites (as shown with dashed border in Fig. 2), and set periodic boundary conditions for the underlying tilted square lattice with $L_x = L_y = L$. States which satisfy the Rydberg constraint can be represented as hard-core dimer coverings; examples of the same are shown in Fig. 2 for

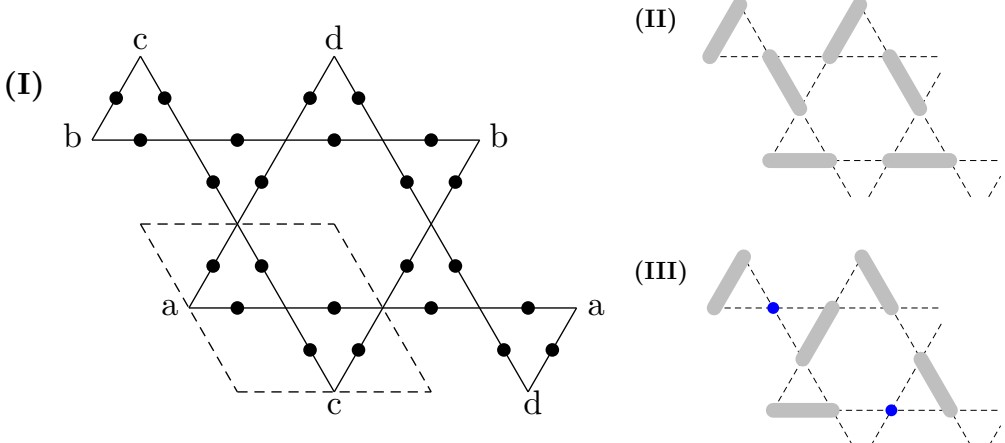

Figure 2: (I) Kagomé link lattice with sites (highlighted) living on the links. A single unit cell has 6 sites and is shown with a dashed line border. Periodic boundary conditions are indicated with pairs of alphabets. (II) Sites occupied by bosons represented by a fully packed hard-core dimer configuration which respects the Rydberg constraint. (III) Partially filled dimer configuration with some sites on the Kagomé lattice (shown with blue filled circles) not covered by dimers.

both the fully filled and partially filled cases.

We begin here with a description of the diagonal update, which does not modify the rod configuration underlying the operator string, but performs the crucial insertions and deletions of diagonal operators. This is followed by three cluster updates, which modify both the rod configuration and the operator string, and they are presented in increasing order of spatio-temporal range. We end by discussing the optimized parallel tempering method used to ensure ergodicity at low temperatures. For all updates other than the diagonal one, we present benchmarks for efficiency. As we will exclusively study the Kagomé link lattice from this point onwards, we will use dimer density to refer to link occupation (rather than boson density as done above).

## 3.1 Diagonal update

A particular configuration during our Monte Carlo simulation is given by a fixed $s_n$, and fixed set $\{\alpha_1, ..., \alpha_M\}$. The diagonal update for SSE [30] changes the configuration by only modifying the operators in $s_n$ which are diagonal. This is a local operation which does not require any modification of operators other than the one under consideration, or of any basis state $\alpha$. As the modifications do not depend on the status of other operators, we can carry out this update by sequentially running once over the entire operator string. The updating procedure is as follows.

We pick a particular location in our fixed length operator string. If this location is occupied by a diagonal operator, we attempt to remove this operator with probability

$$\frac{M - n + 1}{\beta N(\delta + \frac{\Omega}{2})}. \tag{3}$$

If this is greater than unity, then we interpret it as probability 1. This follows in a straightforward manner from the detailed balance condition, and has been explained in detail for the Heisenberg model in Ref. [30] and for the Rydberg model in Ref. [24].

If the location in the operator string which we have picked is empty, we choose to fill it with an operator with probability

$$\frac{\beta N (\delta + \frac{\Omega}{2})}{M - n}. \tag{4}$$

If the decision to add an operator is accepted, we choose to pick an identity operator with probability $P_I = (\Omega/2)/(\delta + \Omega/2)$, or an $H_V$ operator with probability $1 - P_I$. Once an operator is chosen it is assigned to a random site on the lattice. If we chose an $H_V$ operator, then we check if the local background at the chosen site is consistent with the pattern required by $H_V$ (as shown in Eq. (2)), and if so, only then we add this operator. If an identity operator is chosen, there is no such condition, and we always add it. The procedure discussed here closely follows the diagonal updates introduced for the SSE framework for the Heisenberg model.

## 3.2 Local segment update

This update can be understood in the rod language as local deletions/insertions of rod segments. These segments are generically expected to be short in the imaginary time direction for most of parameter space, and thus this update is local in space (acting only on a single site) and local in time (modifies the state of the corresponding site on only a few slices). This is implemented in two ways, as described in App. A, and shown schematically using representative examples in Fig. 3(a).

Without the Rydberg constraint, this update would be identical to the single line updating step for the transverse field Ising model [27]. However, as our model is defined only within the Hilbert space where the Rydberg constraint is not violated, introducing a new rod segment can only be done if it is not adjacent to another rod at the same slice in imaginary time. This is illustrated in Fig. 3(a), where rod position 2 is a valid insertion, whereas position 3 is not due to an overlap with its neighbor. This check does not need to be performed while removing a rod as the Rydberg constraint only prohibits the existence of neighboring rods and puts no constraint on the position of absent rods.

At high temperatures, the operator strings are short and can be intuitively understood to be nearly absent of operators. In this limit, we still need an update which is able to sample the entire Rydberg allowed state space. This is done by picking a site at random and checking that the entire column corresponding to it is operator free. If so, we flip the status of this column after ensuring that there is no conflict created by this flip. This mechanism trivially ensures completely ergodicity at infinite temperature.

To gain an intuition of the efficiency of this update, let us consider the classical limit $\delta \gg \Omega$. In this limit, as we have already discussed the rod configuration is dominated by long rods that break spatial symmetry. In this limit, the rod coverage is well defined and incompressible, i.e. the configurations which dominate the partition function have almost identical rod density, which is the highest attainable without violating the Rydberg constraint. This implies that any update which changes rod density is unlikely to succeed due to the small probability of the configurations it creates. In the opposite limit, $\delta \ll \Omega$, the rod configurations have large numbers of rod ends and no preference for rod coverage, leading to high probabilities for our local segment update. This can also be directly understood by studying the average compressibility, $\kappa = \langle n_i^2 \rangle - \langle n_i \rangle^2$, as a function of $\Omega$ for fixed $\delta$, and indeed it is known that for the Kagomé link lattice [1], regions of large compressibility are present at intermediate $\delta/\Omega$ and this is where we find that our update is highly successful.

To ensure that the update samples over the entire operator string on average in each Monte Carlo step, we track the average area of space time that the update covers on each

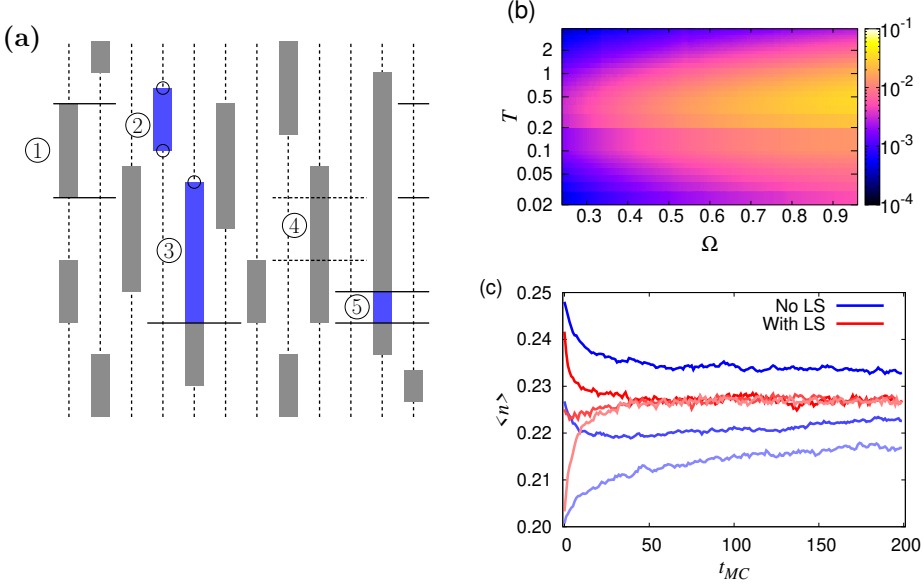

Figure 3: (a) Illustration of the possible moves which can be engineered within the local segment update : 1. A rod segment bounded by two $H_K$ operators can be removed and the operators replaced by $I$ operators (represented as circles); 2. The inverse of process 1, the empty region targeted is highlighted in blue; 3. An empty region, between an $H_K$ and $I$ operator, which cannot be flipped as it would conflict with a nearest neighbor rod; 4. Rod segment between two $H_V$ operators can be flipped to an empty space bounded by two $H_K$ operators; 5. Inverse of process 4. (b) Average success probability of the local segment update. (c) Decay of dimer density ($\langle n \rangle$) as a function of Monte Carlo time for three different initial conditions with and without the local segment update ($L = 4, \Omega = 0.6, T = 0.15$). Error bars are of the order of the temporal fluctuations and are suppressed for clarity.

attempt, and choose the number of repetitions of the update to cover all of space time. We can also track the total size of the regions that are updated successfully. Although these regions can overlap as each initial position is chosen randomly, this size as a fraction of the full space-time can be taken as a proxy for the success probability. This gives us a metric for the efficiency of our update, and our results are shown for a range of temperature and $\Omega$ in Fig. 3(b). Though this update is expected to be highly successful in the $\Omega \gg \delta$ regime, we still find a small but sufficient success rate for intermediate $\Omega$ and $T$. In the limit of $\delta = 0$ and finite $\Omega$, a similar local update [33] has been used to successfully study the PXP model on the square lattice [34]. As argued from the compressibility, we find that the success rate is much smaller in the frustrated regime ($\delta \gg \Omega$). This is seen especially clearly at low temperatures, which are required to understand genuine quantum many-body effects.

As the acceptance rate is not necessarily an indicator of good sampling for the observables of interest, we study the correlation in simulation time of the dimer density (rod coverage discussed above) with and without this update, in the presence of all other updates. We choose this observable as inserting and deleting rods directly changes $n_i$, which none of the other updates do explicitly. We choose $T = 0.15$ and $\Omega = 0.6$, as

Figs. 3(b), 4(b) and 6(b) show that the acceptance probability for all three off-diagonal updates which we have discussed is close to maximum here, and study the total dimer density at a fixed imaginary time slice ($n = \frac{1}{N} \sum_i n_i$) starting from various initial conditions. The latter are generated using simulations which employ all updates and are thus representative of the equilibrium distribution. We deliberately do not use parallel tempering here as we want to follow the loss of memory of the initial condition using just the updates available for individual copies of the system. In Fig. 3(c) we show that without the local update, the dimer density does not equilibrate to a unique value as a function of Monte Carlo time for three different initial conditions when averaged over many realizations of the Markov chain dynamics. This implies that the simulation is not ergodic. However, the addition of the local update leads to the a single equilibrium value ,as seen in Fig. 3(c), for the same initial conditions, with a relatively short Monte Carlo time duration required to lose memory of the initial condition. The data presented here is for $L = 4$, and the same behavior is expected to hold at higher $L$. This clearly demonstrates the importance of this simple update.

## 3.3 Vertical shuffle update

As discussed above, the local segment update is inefficient in the classical limit, where the rod coverage is close to maximal, and creation/annihilation of segments is strongly suppressed. The rod configurations in this limit ($\Omega \ll \delta$) can be visualized as long rods which individually span either the entire time direction or a large fraction of it, thus fixing the same pattern for boson occupation for a large number of time slices, with few uncorrelated segment breaks intervening. This is illustrated in Fig. 4(a) for the 1D chain, and can be easily visualized for 2D lattices by considering any fully packed spatial arrangement. As shown in Fig. 4(a), the number of $H_V$ type operators far exceeds the number of $H_K$ type in the limit where this update is expected to be efficient. The short segment breaks can now be considered to be free objects which can be shuffled vertically on the same spatial site within regions which do not create rod exclusion violations. This is a micro-canonical update as only the positions of operators is modified, not their type, leaving the probability of the operator string unchanged. For this update, we ignore the identity operators, and build a connected linked list of only the $H_K$ and $H_V$ operators assuming that the identity operators are absent. Note that during this listing process we must exclude segment breaks where the intervening time slices host a rod on one of the spatial neighbors of the chosen site. Such a segment is immobile as replacing it with a rod would create two neighboring rods, which is a violation of the Rydberg constraint. The details of the update process are presented in App. B.

We expect vertical shuffling to be effective in the regime of low temperature (large temporal extent) and weak quantum fluctuations. Our numerical results for the efficiency, calculated as discussed above, are shown in Fig. 4(b) for the Kagomé link lattice, and we find that indeed in the regime discussed above, we have success rates which are $O(1)$. At high temperatures, there are relatively few operators in space-time, thus the required local environment is hardly ever seen by the update. For a particular time slice, the action of this update would be to modify the total dimer occupation one dimer at a time. The dimer density ($n_i$) is an important physical observable and the direct impact of this update on the estimation of $n_i$ can be understood by considering the autocorrelation function with and without the vertical shuffle update. We define the autocorrelation function as

$$A(t) = \sum_\tau \frac{1}{N} \sum_i (\langle n_i(\tau + t) n_i(\tau) \rangle - \langle n \rangle^2), \tag{5}$$

where $t, \tau$ denote simulation time, and $n = \frac{\sum_i n_i}{N}$ is the dimer density. The autocorrelation

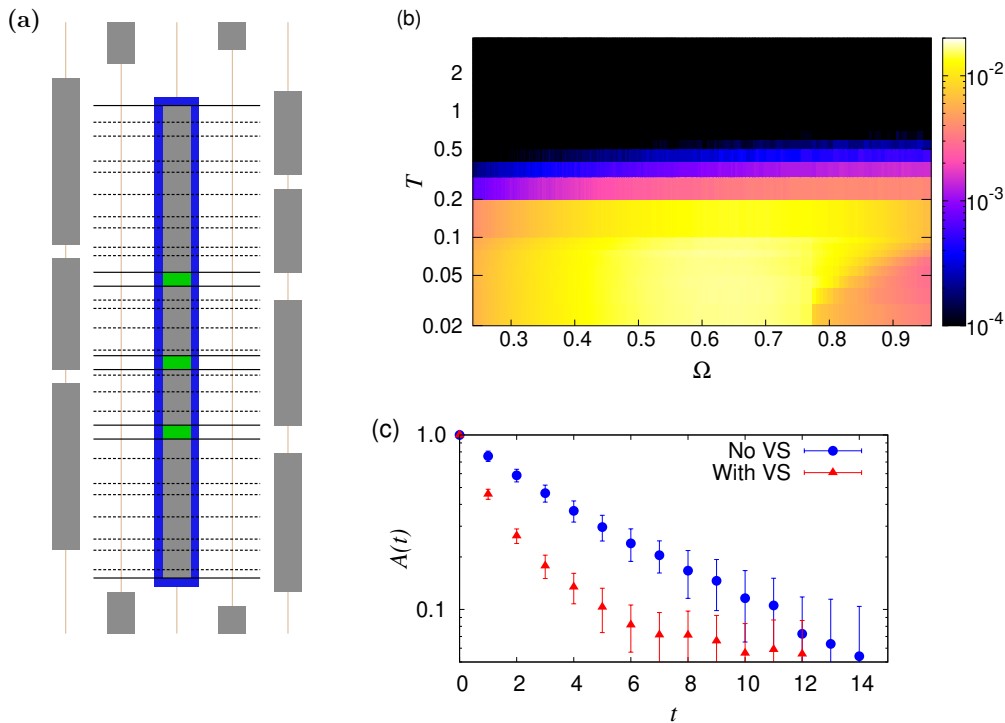

Figure 4: (a) Example of an active region, shown with a blue border, which comprises multiple empty segments bounded by $H_K$ operators. The entire active region for the vertical shuffle update is made up of all active regions of the type shown here which live on the chosen spatial site. (b) The success probability of the vertical shuffle update is seen to be significant only at low temperatures. (c) Autocorrelation of dimer density with and without the vertical shuffle update for $L = 4, \Omega = 0.4$ and $T = 0.06$.

is calculated using the spatial configuration on a random reference slice, and the location of this reference slice is kept fixed for all simulations. We choose $\Omega = 0.4$ and $T = 0.06$, as this is a region where the vertical shuffle update is expected to have a high success probability (Fig. 4(b)) and the local segment update has a low to medium probability (Fig. 3(b)). We show our results for $A(t)$, normalized by $A(0)$ for convenient visualization, in Fig. 4(c), and find that the autocorrelation decays faster when the vertical shuffle update is implemented. This improvement is not drastic as we do not find any region in our phase diagram where we have long rods with many small segment breaks. However, as both the local segment and vertical shuffle updates are local in space-time, the improvement over using just the former is not expected to scale with system size. To see this, let us consider the probability with which the local segment update achieves the $H_K$ pair to $H_V$ pair swap implemented here. Replacing two $H_V$ operators by two of $H_K$ type will be accepted with a probability given by $(\Omega/2)^2$, which is $\ll 1$ in the limit of small $\Omega$, but is system size independent.

## 3.4 Rod diffusion update

The two updates we have described above are limited in their action to a single spatial site. This is highly restrictive and immediately suggests that our simulation would not be able to sample configurations with different spatial patterns efficiently. To remedy this,

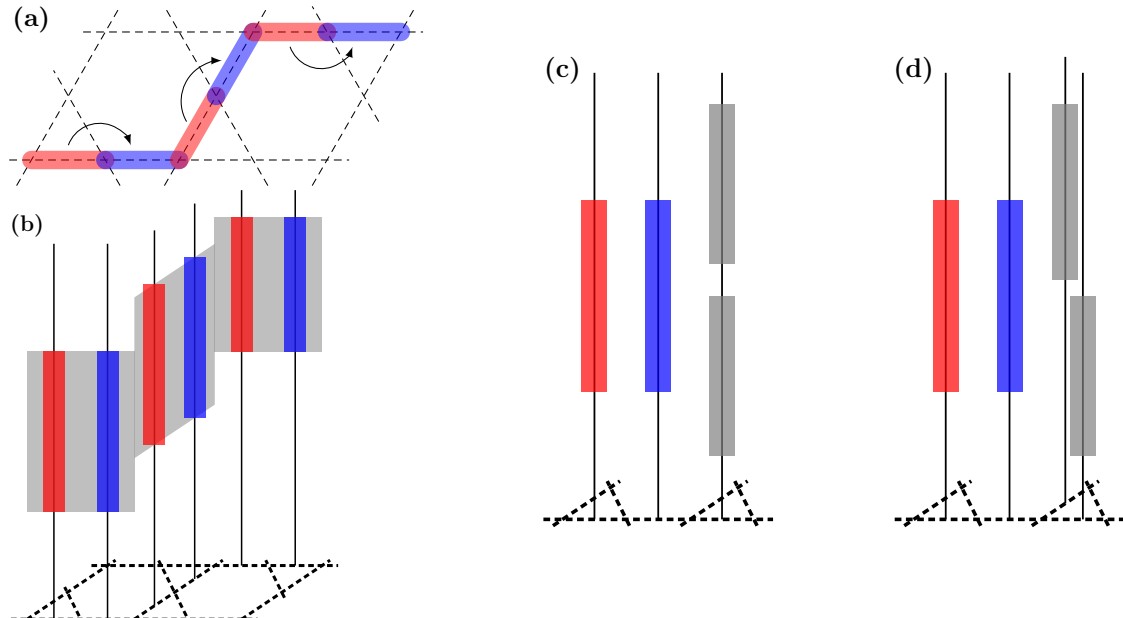

Figure 5: (a) A spatial worm update for a classical dimer model which serves as the template for the corresponding rod diffusion update shown in (b). In all cases, red denotes current position and blue denotes proposed position. In (b), the membrane of modification is shown in gray, and its temporal extent changes with spatial location (rods which do not participate in this update are not shown to avoid clutter). (c) and (d) represent cases where the update must be aborted due to the conflicting number of rods to be moved being $> 1$.

we introduce here a cluster update which moves rods in both space and time. Similar to the vertical shuffle update, this is a micro-canonical update, as we do not change the numbers of $H_V$ or $H_K$ type operators, but only their positions. Identity operators play no role in this update. The detailed balance relations used here are non-trivial only for a spatial system which is at least two-dimensional, and thus we describe this update on the Kagomé link lattice (Fig. 2). This update can be thought of as the worm update for classical dimer models [35], extended into the temporal dimension. Although this may suggest that our update stays in a fixed temporal slice, this is not the case, and it allows for additional flexibility as explained below.

We begin by creating a list of all rods and their positions in the current space-time configuration. This needs to be done only once if the update is to be repeated a set number of times. After each successful update, we just edit the details of the rods which have changed positions. Of all available rods, we pick one at random to initiate the update. We then proceed to sequentially move conflicting rods, thus building a membrane of modified rods as we go, until we reach a configuration which is a valid rod configuration. We use a simple solution to the detailed balance equations which requires that there should be at most one conflicting rod at each step of the membrane building process (Fig. 5(b)). A detailed discussion of this condition as well as of the recursive steps required to move rods is presented in App. C.

To understand the impact of this update intuitively, let us first consider the limit of large $\Omega$. In this limit, the rod coverage is low, and we do not expect a dense packing of rods. Rod movements will statistically encounter no conflicting arrangements, and the update will terminate within a small number of moves. In the opposite limit of zero $\Omega$, rods do

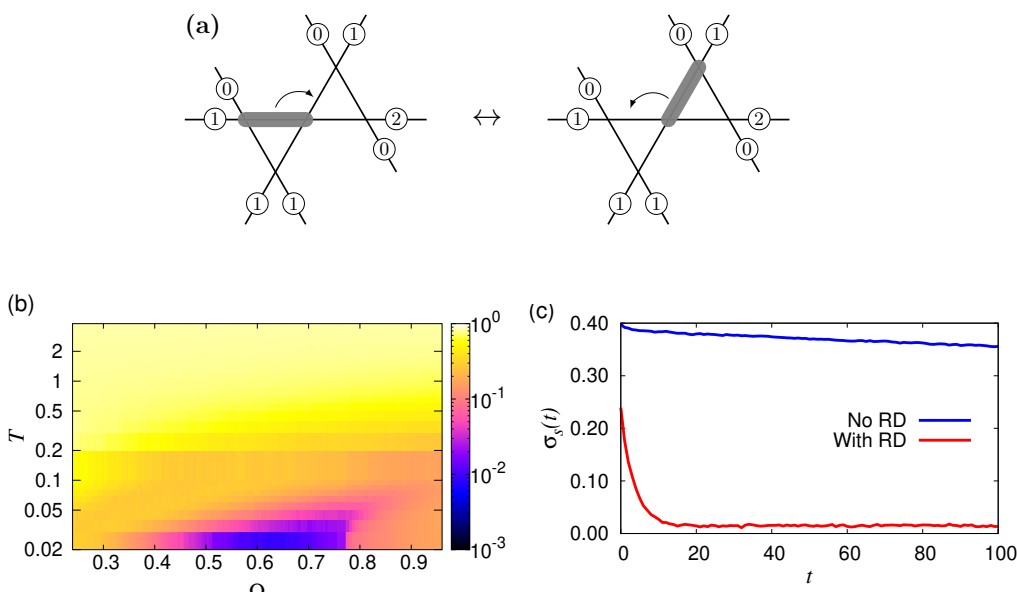

Figure 6: (a) A single move within the rod diffusion update shown by projecting it onto the spatial plane. Circled are the numbers of rods at each spatial site which have a temporal overlap with the rod being moved (shown as gray dimer). This move can proceed as proposed as the number of legal forward options ($= 1$), is the same as the number of legal backward options ($= 1$). (b) The success probability of the rod diffusion update for an $L = 8$ lattice. (c) Spatial variance of the dimer density (Eq. (6)) as a function of Monte Carlo time with and without the rod diffusion update for $L = 4, T = 0.2$ and $\Omega = 0.6$.

not have any end points, and the condition of more than one conflicting rod is never met as every spatial site is occupied by at most one time-spanning rod, and nearest neighbors cannot be simultaneously occupied (as in Fig. 5(d)). The space-time configuration can thus be collapsed to only a spatial configuration, which is the natural compression of the path integral in the absence of quantum fluctuations, and considered purely as a dimer covering. Our update thus reduces exactly to a worm update for a classical dimer model, and has a success probability of unity in this limit. This implies that for the two easily understandable limits, we can expect the update to have a high success probability, but in the intervening region this probability may be suppressed due to high incidences of having strictly more than one conflicting rod in all possible directions of movement. This would correspond to highly jammed configurations which cannot be easily updated.

As explained for the previously described updates, we estimate the region of space-time visited by a single update on average without regarding whether the update is successful or not, and use this estimate to calculate the number of repetitions required to fully sample space-time. Now we can estimate the fraction of space-time which is successfully updated in one Monte Carlo step by our update, and use this to understand the efficiency of the update. We show numerical results for the same in Fig. 6(b) for a range of temperature $T$ and $\Omega/\delta$, and we find that indeed in the two limits discussed above and for values of temperature $\ll \delta, \Omega$, our update has an $O(1)$ success probability. As shown in Fig. 6(b), the inefficiencies begin to be visible only at low temperatures and intermediate values of $\Omega$.

This update is critical for changing spatial patterns in configuration space. To highlight

the importance of this, we can study the decay of a random fixed pattern chosen at equilibrium, and then study the equilibration process with and without this update. We do this by considering the spatial variance, defined as

$$\sigma_s^2(t) = \frac{1}{N} \sum_i (\langle n_i \rangle_{av} - \bar{n})^2, \tag{6}$$

with $\bar{n} = \frac{1}{N} \sum_i \langle n_i \rangle_{av}$, where $av$ denotes averaging over a large number of Markov chains. This is calculated as a function of Monte Carlo time. To analyze the efficiency of this update, we choose $\Omega = 0.6$ and $T = 0.2$, as this is choice of parameters belongs to a regime where the rod diffusion update has a high relative probability (Fig. 6(b)), and the other updates are not efficient (seen in acceptance probabilities in Figs. 3(b),4(b)). Our results are presented in Fig. 6(c), and we see that the local updates are not able to lose memory of the initial condition as the variance does not decay significantly, whereas using the rod diffusion update the variance drops to zero rapidly. The small residual value of the variance is due to the finite number of Markov chains which we have used (as $\langle n_i \rangle_{av} = \bar{n}$ only in the limit of averaging over an infinite number of Markov chains).

An extension of the worm update to spatio-temporal regions which do not contain quantum fluctuations has been implemented for the quantum dimer model [36]. Additionally, a different version of the worm update (where the update sweeps across all of imaginary time without any breaks) has recently been utilized for the Rydberg atom array on the Kagomé lattice [37]. These updates are useful at intermediate temperatures and are closely related to the rod diffusion update discussed above. Toggling of rods on local motifs (such as pairs of sites) has also been used for a frustrated transverse field Ising model [38], and this would again be accessible as a space-local version of our updates.

## 3.5   Optimized parallel tempering

As the updates described above work efficiently in different regions of phase space, it is important to engineer a way to mix configurations from simulations at different parameter values. One of the most commonly used ways to do this is by parallel tempering [39], where simulations (replicas) are simultaneously run at slightly separated parameter values, and space-time configurations are exchanged using the detailed balance condition

$$\frac{P(\{A_1, A_2\} \rightarrow \{A_2, A_1\})}{P(\{A_2, A_1\} \rightarrow \{A_1, A_2\})} = \frac{P_1(A_2)P_2(A_1)}{P_1(A_1)P_2(A_2)}, \tag{7}$$

where $P_{1(2)}$ corresponds to the probability distribution and $A_{1(2)}$ to the current operator string configuration at parameter set 1(2). It is desirable to have a parameter spacing such that configurations living in the limits of phase space where we expect our updates to lead to fully ergodic behavior are able to exchange their positions, thus leading to a high degree of mixing at all intermediate parameter values. However, as system size is increased, the spacing between neighboring parameter sets needs to be reduced to ensure a high rate of exchange from the detailed balance condition. This requires an increasing number of parameter values at which simulations must be run, leading to limitations on size due to computational resources.

It is known from free energy considerations [39] that in regions of parameter space where free energy varies slowly, it is possible to have a good exchange rate even if parameter spacing is large. Small spacing is thus only required close to phase transitions. Following Ref. [40], we use a feedback optimized mechanism to choose the parameter spacing to ensure that we have even diffusivity of the replicas at all points in phase space. Our optimization procedure is carried out at a fixed $T$ and varying $\Omega$ for $\delta = 1$. We fix an

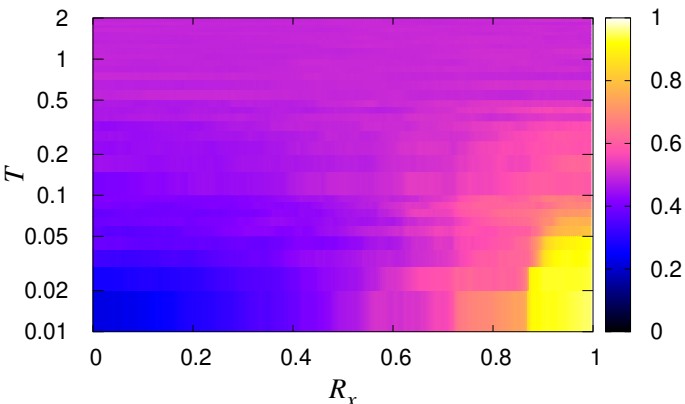

Figure 7: The average replica number at each point in parameter space (replica index $R_x$ as defined in the text). The replica index is normalized to be between 0 and 1. At high temperature we see that at each location, all replicas appear over the course of the entire simulation. The data shown above has been obtained for an $L = 4$ system.

upper and lower value for $\Omega$ such that we are sure that our updates will provide good ergodicity in these two limits, and choose 254 intervening values $\Omega_i$, whose locations can be optimized. This requires a total of 256 parallel simulations, and this number is chosen as the computer hardware at our disposal allows efficient parallelization over 256 cores for each node. The success of the optimization procedure can be quantified by measuring the fraction of replicas which have most recently visited one of the ends of the parameter range. In practice we assign a label "up"("down") to replicas which have most recently visited the upper(lower) end. At each parameter value, we calculate histograms $n_{up}$ and $n_{down}$, which quantify the number of replicas which lived on this parameter value with each label. For convenience, one can now express the number of replicas with the down label as a fraction,

$$f(\Omega_i) = \frac{n_{down}(\Omega_i)}{n_{up}(\Omega_i) + n_{down}(\Omega_i)}. \tag{8}$$

The locations of $\Omega_i$ are chosen to maximize the current of replicas, and if the optimization procedure is successful, one expects $f(\Omega_{i+1}) - f(\Omega_i)$ to be constant for all $i$. Although this ideal scenario is not reached in our optimization, we find that the variations are small, and that at all values of $\Omega_i$, we have replicas which travel to both ends of the parameter range. This is sufficient to ensure ergodicity.

For the parameter range we have picked, we exchange nearest neighbor pairs, i.e. replicas living at $\Omega_i$ and $\Omega_{i+1}$. At each Monte Carlo step, we alternate between exchanging replicas amongst odd or even numbered pairs. This implies that at a particular simulation step, we attempt to exchange $A_1 \leftrightarrow A_2, A_3 \leftrightarrow A_4, ...$, and on the next step, we attempt $A_2 \leftrightarrow A_3, A_4 \leftrightarrow A_5, ....$ We can study the efficiency of our optimization by tracking how far replicas are able to move within a set simulation time. We denote the initial location of the replica by $R_x$. This would naively be an integer between 0 and 255, and we normalize it to be in the interval $[0, 1]$. We calculate the average location of each replica over the entire duration of a simulation, and plot it as a function of $R_x$. Our results are shown for a range of $T$ in Fig. 7, and we see that at high temperatures, the average location of every replica is close to 0.5, and only at $T < \Omega_{mean}/10$, the average location is correlated

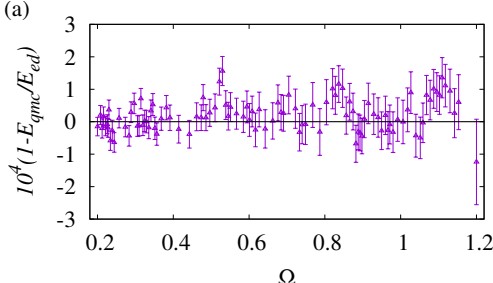
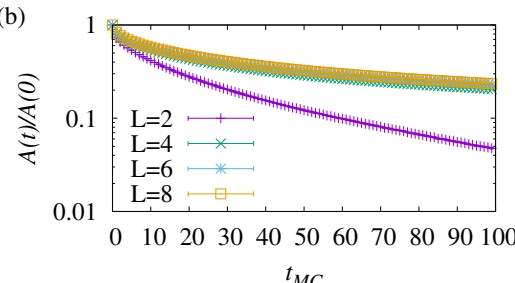

Figure 8: (a) Relative difference between energies using exact diagonalization and QMC. The data has been obtained using $L = 2$ and $T = 0.006$, and the difference is amplified by a factor of $10^4$ to show agreement within error bars. (b) Autocorrelation function as defined in Eq. (9), as a function of Monte Carlo time, for various system sizes at $T = 0.05$.

with the initial position. The data presented in Fig. 7 is for $L = 4$, and using the limits for $\Omega$ as 0.24 and 1.00. For these values, we have already shown in Fig. 6 that we have high efficiency for the rod diffusion update. Note that the optimized parallel tempering which we have discussed above is not efficient at low temperatures, and thus we need updates which are able to sample efficiently in each regime separately. Thus, using just a combination of parallel tempering and simple local updates would be insufficient at low temperatures.

### 3.6 Benchmarking and autocorrelation functions

To ensure that we have implemented the detailed balance conditions correctly, we perform a comparison with exact diagonalization (ED) for a small system size. For the 24-site periodic lattice shown in Fig. 2, the number of dimer occupation states allowed by the Rydberg constraint are 2649. This number is small enough to perform a full diagonalization, and thus to calculate energy at finite temperature. We carry out a comparison of the energy for a range of $\Omega$ as shown in Fig. 8(a), and find that the agreement between ED and QMC is of the order of 0.01%, and within the error bars of the QMC simulation. Thus we can conclude that the QMC simulations are unbiased and accurate. Note that in Fig. 8(a), we have used $T = 0.006$, which is $\ll \Omega$ for the range of $\Omega$ studied.

As energy is a coarse grained observable, it is also important to study correlations of the space-time structure to understand how effective our updates are modifying the same.

We do so by considering auto-correlations of the dimer occupation, defined as

$$A(t) = \sum_{\tau} \frac{1}{N} \sum_{i} \langle n_i(\tau + t) n_i(\tau) \rangle - \langle n \rangle^2 \,, \tag{9}$$

where $t, \tau$ denote simulation time, and $n = \frac{\sum_i n_i}{N}$ is the average dimer density. The autocorrelation is calculated using the spatial configuration on a random reference slice, and the location of this reference slice is kept fixed for all simulations.

As we are running a large number of replicas in parallel, and exchanging them at every Monte Carlo step, we calculate the autocorrelation for each replica, by just considering its spatial pattern. Thus we observe how the pattern de-correlates as it moves over parameter space. As we expect all replicas to visit all of parameter space, we average over the autocorrelation functions of individual replicas, and thus improve the associated statistics.

The data for the autocorrelation is only collected after the simulation has had a sufficient number of warm up steps, and is sampling from the equilibrium distribution. As we would like to know the performance of our algorithm at low temperature and intermediate values of $\Omega$, we fix $T = 0.05$ and $\Omega \in [0.24, 1.00]$. The autocorrelation for various system sizes is shown in Fig. 8(b), and we see that $A(t)/A(0)$ decays rapidly for $L = 2$, and saturates for the 3 larger system sizes, namely $L = 4, 6, 8$. This result suggests that the algorithm is fairly efficient at sampling even for $T \ll \Omega$ for the most challenging range of $\Omega$ expected.

# 4 Phase diagram of Kagomé link lattice

We now study the Kagomé link lattice using ED for the 24-site periodic lattice to gain an understanding of the phase diagram and study larger system sizes using the QMC updates discussed in the previous section. The latter allow us to extract the phase diagram in the thermodynamic limit. The Rydberg atom array on this lattice has been discussed in Ref. [1], where it was found that in the limit of $\Omega \gg \delta$, the ground state is a trivial paramagnet in the language of spin-1/2 particles, and that by lowering $\Omega$, one can cross into a spin liquid phase and eventually into a valence bond solid phase with a large unit cell. These conclusions were reached by carrying out DMRG simulations on quasi-1D cylindrical geometries, and we would like to study the temperature dependence using our QMC directly in the 2D limit. The effect of temperature is most easily understood in the classical limit, where at high temperature, the system accesses all Rydberg allowed configurations, and at $\beta \gg \delta$, the system only accesses the manifold of states which have maximal density. This set is still macroscopically degenerate due to the inherent frustration of the underlying lattice. Once a quantum fluctuation is added, we expect a second energy scale controlled by both $\Omega$ and $\delta$, where the residual classical entropy is released, leading to a ground state with at most $O(1)$ degeneracy.

Using ED, we study the specific heat as a function of both $\Omega$ and $\beta$ (for convenience we set $\delta = 1$). We can trace the lower release in entropy as a peak in the specific heat, and we show this in Fig. 9(a). We find that this peak moves smoothly from $T \approx 10^{-5}$ at $\Omega = 0.3$ to temperatures of $O(1)$ at high $\Omega$. This suggests that the curve traced by the location of the peak in Fig. 9(a) ends at $T = 0$ for $\Omega = 0$, which would be the natural expectation. At $\Omega \approx 1$, the distinction between the two peaks in specific heat which signify the release in entropy due to classical constraints and quantum fluctuations is lost. This is expected for $\Omega \approx \delta$ as the separation between scales required for two distinct peaks is lost. This phase diagram suggests that the quantum paramagnet is the correct ground state for all $\Omega > 0$. This is in contrast with the expectation of a spin liquid phase. We find that the low lying spectrum for small $\Omega$ has a unique ground state with a roughly equally spaced spectrum above it. This is also inconsistent with a $Z_2$ spin liquid on a torus, where one would expect four low lying states which have a topological nature, and are well separated from the rest of the spectrum.

The QMC simulation setup can now be used to extract the residual entropy as a function of temperature and $\Omega$. As our simulations are restricted to sample only from the blockade allowed states, the thermal entropy at infinite temperature cannot be easily calculated from product states, as dimer coverings cannot be represented in such a manner. To calculate a benchmark for the infinite temperature entropy, we use specific heat data at large $\Omega(\approx 1.4)$, and integrate $C_v/T$ starting from temperatures much below the expected gap. As the system is in the paramagnetic phase in this limit, the ground state residual entropy is trivially expected to be zero, and the integration to high temperatures allows us to extract the infinite temperature entropy. We find that the entropy per unit site is

(a)

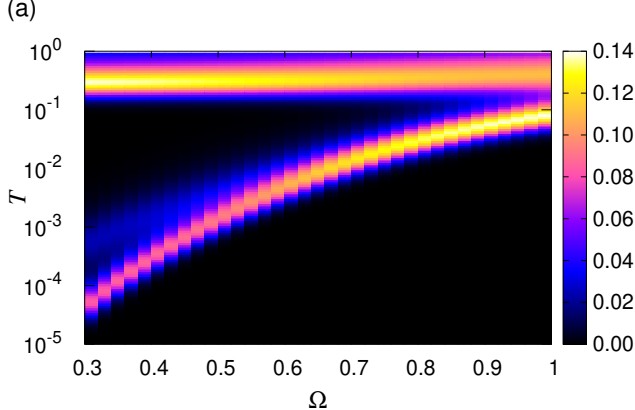

(b)                             (b)

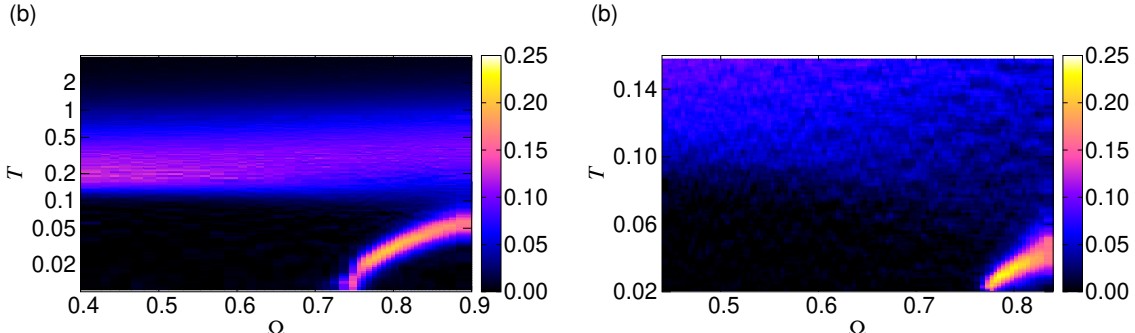

Figure 9: Specific heat for the Kagomé link lattice : (a) the $L = 2$ case, simulated using exact diagonalization, shows a smooth shift of the lower peak with $\Omega$; (b) the $L = 4$ case with QMC, where temperatures as low as 0.01 can be reached, shows a behavior which is qualitatively consistent with the $L = 2$ case. (c) the $L = 8$ case, again similar to the $L = 4$ case.

0.3285(1) for $L = 4$, and has negligible finite size corrections for larger sizes. As expected, this is significantly smaller than $ln(2)$, which would be the case without the Rydberg constraint.

We show our results in Fig. 9(b) and (c), and find that these are similar to the 24-site lattice for system sizes of 4 and 8 down to temperatures of $\approx 0.01$. For low temperatures, calculating $C_v$ using the fluctuation- dissipation relationship, i.e. $C_v = \beta^2(\langle E^2 \rangle - \langle E \rangle^2)$, has large error bars, and thus we obtain $C_v$ directly by taking a numerical derivative of the energy. For our simulation ranges, this ensures an error bar of $< 10\%$. The smallest value of $\Omega$ at which this temperature is low enough to detect the second peak in specific heat is 0.7, and the temperature considered is 30 times smaller than the smallest scale in the system ($\Omega/2 = 0.35$). This point is consistent with the location of the transition to the spin liquid phase in Ref. [1]. If the spin liquid phase indeed appears close to this transition, it is not seen down to temperatures much smaller than $\Omega, \delta$.

To show that the low temperature phase at small $\Omega$ which we see in our simulations is a classical spin liquid, we use the residual entropy and correlation functions. The former is shown in Fig. 10(a) for $\Omega = 0.4$ and 0.9, and calculated using the reference $S(T \to \infty) = 0.3285(1)$ as discussed above. We see that at low temperatures for $\Omega = 0.4$, we have a residual entropy, which should eventually drop to zero below a second crossover

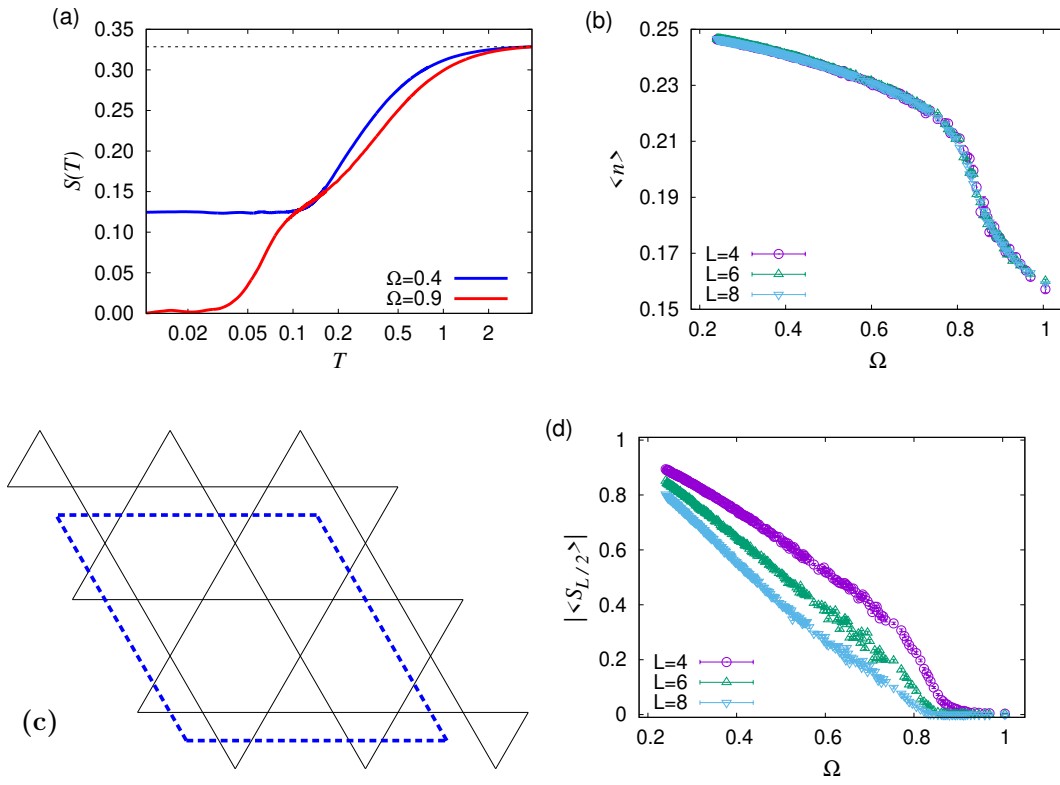

Figure 10: (a) Entropy for the $L = 4$ case, calculated using specific heat data of Fig. 9, for $\Omega = 0.2$ and 0.45. Expected infinite temperature entropy shown by dashed line. (b) Dimer density $< n >$ as a function of $\Omega$ for $L = 4, 6$ and 8 shows smooth behavior at $T = 0.05$. (c) Contour for calculating string order parameter $S_{L/2}$ for $L = 3$. (d) $S_{L/2}$ as a function of $\Omega$ for $L = 4, 6$ and 8 at $T = 0.05$.

temperature which is too small to access currently. We define a dimer correlation function as

$$C(r) = \frac{1}{L^2} \sum_{i_x, i_y} \sum_{d=1}^{6} (\langle n^d_{i_x, i_y} n^d_{i_x+r, i_y+r} \rangle - \langle n \rangle^2), \tag{10}$$

where $d$ indexes the position within a unit cell, and $i_x$ and $i_y$ the position of the unit cell. For the classical dimer model on the Kagomé lattice, this correlation function is known to be exactly zero past nearest neighbors [32]. As the classical spin liquid in our model can be thought of as the same with monomers, we expect that we should have an exponential decay of $C(r)$ and we find that this is indeed the case in our simulations, where we find $C(r)$ vanishes up to an error bar of $10^{-3}$ for all $r$.

The crossover between the spin liquid phase and the quantum paramagnet can be captured using the dimer density ($\langle n \rangle$) and indirect observables such as string order parameters. We fix $T = 0.05$ for our analysis and first show our results for $\langle n \rangle$ as a function of $\Omega$ in Fig. 10(b). We find a rapid decline in dimer density as the ground state crosses from the spin liquid to the paramagnetic regime. However, this decline is smooth and stays unchanged with increasing system size (Fig. 10(b)). Note that although this is expected for a crossover, this should sharpen into a non-analytic point for a transition, as seen in the DMRG data of Ref. [1]. To understand the spin liquid behavior better, we now turn to string order parameters. As our simulations are carried out in the dimer occupation basis, we have easy access to the $P$ order parameter of Ref. [1], and use the contour shown

in Fig. 10(c) (this is the same as Fig.6 of Ref. [1]). $S_r$ is simply defined as the product of $f(l)$ along all links ($l$) cut by the contour, where we take $f(l) = -1$ if $l$ is occupied by a dimer and 1 if not, and the contour has size $r \times r$. For a fully packed dimer configuration, $|\langle S_r \rangle| = 1$, where the absolute value is taken to negate the sign oscillation due to odd or even $r$. However, since we have a non-zero monomer density at all $\Omega$, $\lim_{r \to \infty} \langle S_r \rangle \to 0$ necessarily in both phases. For small $r$, we can still expect non-zero values in the spin liquid phase and this can act as a marker for the crossover to the quantum paramagnet. This motivates us to study $\langle S_r \rangle$ for $r = L/2$ for $L = 4, 6, 8$ systems at $T = 0.05$ and we show our results in Fig. 10(d). We see that these order parameter take $O(1)$ values in the spin liquid phase and rapidly decay around the crossover. As expected, $\langle S_r \rangle$ decays with increasing $r$ even in the CSL regime.

These studies do not deny the existence of a spin liquid phase, and although confirming this would require arbitrarily low temperatures, our results show that the assumed gap to this spin liquid is smaller than $10^{-2}$ of the microscopic length scales. This implies that thermal fluctuations will play an important role when attempting to realize this spin liquid in a Rydberg atom array.

# 5 Conclusion

We have presented a construction of cluster updates for stochastic series expansion QMC for Rydberg atom arrays using a mapping to a hard rod model. The mapping involves vertical rods with an exclusion radius, and is applicable for Rydberg atom arrays on any lattice. These updates are designed to work efficiently in different parameter regimes, and we combine them together using an optimized parallel tempering mechanism. We have studied the efficiency of various updates individually and also considered their combined capabilities by studying the autocorrelation of spatial occupation profiles. Previous attempts to design cluster updates for Rydberg systems have relied on a mapping to quantum dimer models [20, 23], whereas our updates do not have such a constraint and are expected to be able to sample in varying parameter regimes.

The algorithm is applied to study the Kagomé link lattice, where a gapped $Z_2$ spin liquid is expected at low temperatures. Our analysis of the specific heat provides an upper bound on the gap. This bound is estimated to be $< \Omega/50$, for the range of the coefficient of the quantum fluctuation, $\Omega$, where the spin liquid is expected to exist. This implies that the phase is highly sensitive to thermal fluctuations. As our results are preliminary, future studies will involve studying lower temperatures, and other lattices with similar frustration properties which may lead to more robust spin liquid ground states.

The rod model can also independently be analyzed from the perspective of renormalization group to understand analytically the physics of Rydberg systems. There are numerous examples of interesting phases being realized in such systems, and particular manifestations which can directly be analyzed using our approach are a realization of $Z_N$ symmetry in coupled Rydberg chains [41] and a spin glass realized without disorder [23]. The sweeping cluster update has been used to simulate a paradigmatic example of a $Z_2$ spin liquid, namely the toric code [42], and our rod diffusion update can be thought of as carrying out the same task within the Rydberg constrained space. This is shown explicitly in the classical limit, where we recover the worm update which samples between configurations of the classical dimer model, and which are precisely the low energy excitations which one would expect to be generated by the quantum fluctuations. This suggests that this update can also be adapted to $SO(N)$ systems on frustrated lattices, which are expected [43] to host a stable $Z_2$ spin liquid for large $N$.

## Acknowledgements

We would like to thank Owen Benton for numerous discussions, and Fabien Alet and Kedar Damle for pointing out relevant references. Computational resources for this project were provided by MPIPKS. The formatting of the pseudo-code was done following Ref. [44].

## A  Details of local segment update

Following the brief description of this update in Sec. 3.2 and the representative schematic shown in Fig. 3(a), we present the details and a pseudo-code for the same in this appendix.

First we have a micro-canonical update, which switches between two space-time configurations which have the same probability. This proceeds by first picking a random spatial site and a random slice and searching sequentially in slices above and below the chosen one, until we encounter slices in both directions which have either an operator on the site of interest. In the time direction, if we isolate the spatial site of interest and look at only the operators living on this site, then these operators are next to each other in the imaginary time direction. If either of them is an $H_V$ operator, the update is aborted. For comparable values of $\delta$ and $\Omega$, we would expect that the update would reach this stage with a probability of $O(1)$. Once this pair of neighboring operators is identified, we identify if the state of the site for the time region between these two end points hosts a rod or not. If it does, then we can flip this segment with unit probability as we exchange only between $H_K$ and identity, and both have the same coefficient in the Hamiltonian. However, if the segment does not host a rod, we must check the status of the neighboring sites to ensure that in the time region of interest, they are not occupied by a rod. If they are, then flipping this segment will create two neighboring rods, which is not allowed by our basis states, and we must abort the move. After this check, if we find that we can occupy the segment with a rod without creating violations of the Rydberg constraint, then we do so with unit probability. These processes are shown with an example in Fig. 3(a), for rods labeled 1-3. Identity ($I$) operators are represented using circles on a single site, as these operators do not have any dependence on the state. Pseudo-code for this process is presented below.

```
__global__ function Local_Segment(int *opr_string)

   int j,k;
   int s_i = random.site();
   int t_i = random.slice();
       k = t_i +1; while(site of opr at k not s_i) {k++;} t_top=k;
       k = t_i -1; while(site of opr at k not s_i) {k--;} t_bottom=k;
   if (opr at t_top = H_V) or (opr at t_bottom = H_V) : abort;
   else :
       k = rod.starting.upwards.from(t_bottom);
       if (k = no rod) :
       (1) toggle operators (H_K <-> I)at t_top and t_bottom;
       (2) update rod.starting.upwards.from(t_bottom);
       else :
           j = check.if.rod.on.neighbors(s_i,t_bottom,t_top);
           if (j = yes) : abort;
           else : repeat (1) and (2);
           end if
       end if
   end if
```

The second process which we use within this update is a canonical move which toggles

between configurations with different probabilities. We again identify operators at a randomly chosen location on either side of a chosen slice, and proceed with the update only if both operators are of either type $H_V$ or of type $H_K$. If so, then we identify whether or not the segment intervening is covered by a rod segment or not. The presence of a rod segment implies that we can flip the status of a segment without checking its neighbors. However, as we are allowing ourselves to toggle only between $H_K$ and $H_V$ as endpoints, if the endpoint is of the form $H_K$, the resulting arrangement (no rod coverage on both sides of an endpoint) is inconsistent with an $H_V$ operator. Thus if we have a rod covering the segment of interest, we require that the endpoints must host $H_V$ operators. If this condition is not met then we abort our update. If not, then we flip the status of the segment and the endpoints with probability $\delta^2/(\Omega/2)^2$. Now turning to the case where the segment does not host a rod segment; the end points now must necessarily be of the $H_K$ type, as $H_V$ operators require rod coverage on both sides. Before flipping we must check if the flip will create inconsistencies with neighboring rods, if not, then we can accept this flip with probability $(\Omega/2)^2/\delta^2$. The operations described here are illustrated with a simple example in Fig. 3(a) with rods labeled 4 and 5. The pseudo-code for this process is almost identical to the one presented above except for the toggling of $H_K \leftrightarrow I$, which now has an acceptance probability associated with it (similar to the standard metropolis update for the classical Ising model).

## B   Details of vertical shuffle update

Here we discuss the implementation of the vertical shuffle update (Sec. 3.3) along with a short pseudo-code for the same.

First we identify a random site, and run through the entire time direction to identify the regions where we can move rod segments and segment breaks freely. We call this as the "active" region and an example is highlighted with a blue border in Fig. 4(a). The entire active region on the site in interest is the union of all regions like the one shown, which are on the same site (this is generated by scanning vertically). In this process, we also create a list of all the segment breaks and a larger list of all neighboring vertical pairs in the vertical direction, both within the active region. Care must be taken in this procedure to also consider the pair which wraps over the periodic boundary condition in the time direction. In addition we also track the number of neighboring $H_V$ pairs, as we will swap these with $H_K$ pairs in this update. Note that a segment break is defined in our context to mean two $H_K$ operators on the spatial site of interest, which are neighbors in the vertical direction and the space between them is not covered by a rod. In addition, none of the time slices contained between these two $H_K$ operators should host a rod on any spatial neighbor of the site under consideration. If both the numbers of $H_K$ pairs (denoted by $k$) and $H_V$ pairs (denoted by $v$) are non-zero, we proceed by picking a random $H_K$ pair and attempting to exchange it with a random $H_V$ pair. Following the convention developed in previous subsections, $H_V$ and $H_K$ are denoted by dashed and solid horizontal lines respectively, and in Fig. 4(a) we see that there are many more of neighboring $H_V$ pairs than $H_K$ pairs. Although both configurations have the same probability, one must be careful while carrying out this exchange as the forward, $A \to B$, and reverse $B \to A$ processes may not have the same proposal probability (here $A$ and $B$ are the configuration before and after the update). In other words, to satisfy the detailed balance condition, $P(A)P(A \to B) = P(B)P(B \to A)$, we must now require $P(A \to B) = P(B \to A)$, as $P(A) = P(B)$. If configuration $A$ has $k_A$ number of $H_K$ pairs and $v_A$ number of $H_V$ pairs then $P(A \to B)$ is given by $(k_A v_A)^{-1}$, as $B$ is defined by picking one out of $k_A v_A$ options.

The simplest way to consistently follow detailed balance now is to require $k_A = k_B$ and $v_A = v_B$. This is engineered as follows.

We pick an $H_K$ pair randomly and check the operators immediately above and below the pair at the same spatial location. If both of these operators are of the $H_V$ type, then we proceed to pick an $H_V$ pair which also has this property. Now that we have identified two pairs which have the same "local" environment, we can proceed with the exchange with unit probability. The particular check which we have used is one of the easiest ways to ensure that the condition of $k_A = k_B$ and $v_A = v_B$ described above is satisfied as the number of $H_K$ and $H_V$ pairs are individually explicitly conserved. An example of all such $H_K$ pairs is shown in Fig. 4(a) in green coloring.

Since this update scans the entire time span of an individual site, simply repeating this update $N$ times covers all of space time on average. For an individual update, the number of pair exchanges attempted is taken to be the number of $H_K$ pairs identified, as this allows each pair to move once on average. The success probability of our update can be quantified by recording the total length of the pairs which are successfully exchanged as a fraction of the time span of the system for a single update. Note that every proposed exchange is not accepted, as the local environment may not satisfy the condition discussed above. A pseudo-code for this update is presented below.

```
__global__ function Vertical_Shuffle(int *opr_string)

int j,k,t_j,t_{j+1};
int s_i = random.site();
int opr.list = filter(opr_string,s_i);
int nbr.opr.list[number.of.nbrs.of.s_i];
for j in number.of.nbrs.of.s_i :
    nbr.opr.list[nbr.number(j,s_i)] = filter(opr_string,nbr.number(j,s_i));
end for
int Hk.list, Hv.list;
for j in size.of(opr.list) :
    if (opr.list[j].type = Hk) and (opr.list[j+1].type = Hk) :
        t_j=opr.list[j].t-position;
        t_{j+1}=opr.list[j+1].t-position;
        k = check.if.rod.on.neighbors(s_i,nbr.opr.list,t_j,t_{j+1});
        if k = 0 : #no conflicting rods
            add(j,Hk.list);
        end if
    else if (opr.list[j].type = Hv) and (opr.list[j+1].type = Hv) :
        add(j,Hv.list);
    end if
end for

int pair_k,pair_v;
int opr_k_above,opr_k_below,opr_v_above,opr_v_below;
for j in size.of(Hk.list) :
    pair_k = random_element(Hk.list); #pair indexed by lower opr location
    opr_k_above = opr.list(pair_k+2); #gets opr above pair
    opr_k_below = opr.list(pair_k-1); #gets opr below pair
    pair_v = random_element(Hv.list);
    opr_v_above = opr.list(pair_v+2); #gets opr above pair
    opr_v_below = opr.list(pair_v-1); #gets opr below pair
    if (opr_k_above.type=Hv) and (opr_k_below.type=Hv) :
        if (opr_v_above.type=Hv) and (opr_v_below.type=Hv) :
            exchange(pair_k,pair_v);
            update(opr.list);
            update(Hk.list);
            update(Hv.list);
        end if
    end if
```

```
end for
```

## C   Details of rod diffusion update

In this appendix, we detail the procedure for implementing the rod diffusion update presented in Sec. 3.4 along with a pseudo-code for the same.

Each spatial site on the Kagomé link lattice can be visualized as a dimer between two neighboring sites on the Kagomé lattice, and it can be moved by pivoting it about one of its ends, as shown in Fig. 5(a). By extending this into the temporal direction, this pivot corresponds to a movement of a rod. The spatial site of the Kagomé lattice about which this pivot is carried out is chosen to not be in the membrane of modification already developed, as this would lead to the update undoing its own moves. This is similar to the spatial worm update [35], where dimers are pivoted around an exit site, which is not the site via which the worm entered the dimer. For the initial rod, this site is chosen at random, as there is no pre-existing membrane. An example of the membrane of modification is shown in Fig. 5(b). For the Kagomé lattice, a pivoting dimer has three possible options for its new position, as shown in Fig. 5(a). To decide which of these options to choose, we consider the corresponding rod configuration. We calculate the number of conflicting rods generated by moving the current rod to each of these three spatial positions while keeping the temporal ends of the rod the same. We choose the position with the least number of conflicting rods, and denote this number by $n_c$. If there are multiple such positions, we store the number of them by denoting them as the number of forward options ($p_f$), and choose one at random with uniform probability. If $n_c > 1$, detailed balance would require a reverse proposed move which coordinates the movement of $n_c$ rods into the relevant positions, leading to the reverse movement of the current rod. This would imply that we would have to engineer an update which branches, and carries movements of multiple rods at the same step of the update. This would significantly complicate the update, and we cannot visualize a simple way to do this currently. An example of this sort of movement is shown in Fig. 5(c) and (d), where the movement of the current rod (shown in red) to its new position (shown in blue) creates a conflict with two rods. As shown, these two rods can be on the same spatial site or on different ones.

Thus if $n_c > 1$, we abort the update. If not, we consider the reverse move, where the rod is being moved from its future position to its current position. In the dimer language, this is now the reverse pivot. Following the same procedure to pick the new position of the rod leads to a calculation of the number of backward options $p_b$. Detailed balance now requires that the probability of the forward process, $1/p_f$, be equal to that of the reverse process, $1/p_b$. This trivially translates to requiring $p_f = p_b$. If this condition is not met, we abort our update. We relax this condition only for the initial and final rod movement, where we require that $p_f^{start} = p_b^{end}$ and $p_b^{start} = p_f^{end}$. A sample pivot process which satisfies detailed balance is shown in Fig. 6(a). If the update is accepted, we update the relevant operators. With each rod movement, we carry all operators ($H_K$ and $H_V$ type) which are living on it to the next position inhabited by the rod. The update terminates when the current rod is in a position where there are no more conflicting rods to move. A short pseudo-code for this update is presented below.

```
__global__ function Rod_Diffusion(int *opr_string)

#Prepare rod configuration from opr_string beforehand
#Copy opr_string to temp_opr_string

int j,k,l;
```

```
int rod_curr = random.rod();
int pivot = random-end(rod_curr.site); #pivot chosen as one of dimer ends
int t_end = top-end(rod_curr);
int b_end = bottom-end(rod_curr);

#recursion begins, track step number as r-step
int r-step=1;
while(1) : #infinite loop broken when valid config met
    for j in nbrs(rod_curr,pivot) : #positions to which the dimer can pivot
        number-of-valid-exits[j]=0; #positions with conflicts<2 added here

        # Below, choose pivot point for j which is other end of dimer j
        temp-pivot=dimer-end(j,1);
        if (temp-pivot=pivot) : temp-pivot=dimer-end(j,2);

        l=0;
        for k in nbrs(j,temp-pivot) : #counting conflicts if rod_curr -> j
            l+=number-of-rods(k,b_end,t_end); #in range b_end to t_end
            possible_exit[j]=rod-number(k,b_end,t_end);
        end for
        if (l<2) :
            number-of-valid-exits[j]++;
        end if
    end for

    p_f=0; #number of possible forward options
    min-conflict=2;
    for j in nbrs(rod_curr,pivot) :
        if (number-of-valid-exits[j]<min-conflict) :
            min-conflict=number-of-valid-exits[j];
        end if
    end for
    #Above we have preferred the exits with zero conflict over one conflict

    for j in nbrs(rod_curr,pivot) :
        if (number-of-valid-exits[j]=min-conflict) : p_f++;
    end for
    if (p_f=0) : abort;

    int choice_forward = random(p_f); #choose out of p_f options

    #Using choice_forward and pivot, now calculate the number
    #of options p_b for the reverse pivot, using same steps
    #as for the forward move

    if (recursion-step=1) :
        p_f_strart=p_f; p_b_start=p_b; #store number of options for start
    end if
    if (number-of-conflicts[choice_forward]=0) :
        p_f_end=p_f; p_b_end=p_b; #same for end
    end if

    if ((p_f=p_b)||(r-step=1)||(number-of-conflicts[choice_forward]=0)) :
        rod_curr=choice_forward;
        update(temp_opr_string,rod_curr,choice_forward); #move operators
            along with rod
    else :
        abort;
    end if

    if (number-of-conflicts[choice_forward]=1) :
        rod_curr=possible-exit[choice_forward];
```

```
        t_end=top-end(rod_curr);
        b_end=bottom-end(rod_curr);
        pivot=dimer_end(rod_curr,1);
        if (pivot in choice_forward): pivot=dimer_end(rod_curr,2);
        #We want the pivot to be away from choice_forward
        #Below, checking detailed balance for start-end
    else if ((p_f_start=p_b_end) and (p_f_end=p_b_start)):
        exit; #update completes as no conflicting rods
    else :
        abort;
    end if
    r-step++;
end while

if (valid exit without abort) : opr_string=temp_opr_string;
```

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
