# Peer review of "Quantum Monte Carlo simulations in the restricted Hilbert space of Rydberg atom arrays"

_SciPost Physics_

## Round 2 · Referee Report · Anonymous (Referee 1) · 2023-10-23

Strengths

1-An unbiased numerical method is presented for constrained quantum many-body Rydberg atom systems that are of current interest in different fields.
2-Several complementary update schemes are presented.
3-Benchmarking and efficiency tests has been performed.
4-Some physical results on the kagome link lattice model are presented.

Weaknesses

1-In the application to the kagome link lattice model, only a single observable (the specific heat) is discussed.
2-For some update schemes, more technical details could be provided (see below).
3-Minor issues in the presentation (see below).

Report

This manuscript presents a quantum Monte Carlo approach to study Rydberg atom arrays by treating them directly within the reduced Hilbert space enforced by the Rydberg blockade. Using this restriction explicitly in the construction of the stochastic series expansion algorithm, the author introduces various local and non-local update schemes. He also provides benchmarking and an efficiency analysis. As an explicit application, the case of the kagome link lattice is discussed, based on specific heat data down to the relevant low temperatures. No indication for a previously proposed quantum spin liquid phase is obtained from these simulations.

The algorithm introduced here is well explained on a non-technical level and the relevant update schemes are motivated and a performance analysis is provided. For most of the updates, a more technical treatment, or a code base, would certainly be useful for readers who actually want to implement these algorithms themselves. In the application to the kagome link lattice model, only a single observable is considered, the specific heat. This analysis could be extended, e.g., by examining also correlation functions.

Overall, this manuscript certainly is valuable in introducing an unbiased QMC algorithm to provide unbiased numerical data on a very timely topic in quantum many-body systems. Indeed, Rydberg atom arrays are realized in recent experiments, and are met with a broad interest in different communities, including frustrated magnetism and quantum computation. I recommend to publish this paper, after a few changes, requested below, have been considered by the author.

Requested changes

1-In Sec. 4 it would be useful to expand beyond the discussion of the specific heat, e.g., by considering relevant correlation functions.
2-I find the sentence "In the time direction, these operators are neighbors..." in Sec. 3.2 confusing. If these operators are not meant to be neighbors to each other, please rephrase this sentence, e.g., by replacing the "are" by "have" or alike. Otherwise, please explain better how these operators are specified.
3-The whole text should be given a careful reading , as it still contains several typos, e.g., "An Extension", "the the", "rydberg" in various references.
4-For the updates in Sec. 3.2-3.4, the author should provide more technical details regarding the actual implementation of the algorithms, e.g., by providing a code base or appropriate pseudo-code.

---

## Round 2 · Referee Report · Anonymous (Referee 2) · 2024-1-3

Strengths

(1) Addresses an important problem
(2) Presents a variety of technical improvements
(3) The implementation is scientifically solid
(4) An important physical result is obtained

Weaknesses

(1) Technical discussions can be difficult to follow
(2) Details of the algorithms are the focus of the paper
(3) The primary metric for efficiency is the acceptance probability
(4) There is only one physical result

Report

In this work, the author addresses the important problem of improving the efficiency of quantum Monte Carlo (QMC) simulations of models of Rydberg arrays, specifically restricting to the subspace of configurations allowed by the Rydberg blockade. Previous attempts at cluster algorithms for Rydberg models relied on a mapping to a quantum dimer model, whereas the present study is more general.

This work is important, since Rydberg models can host a wide variety of exotic physics and, crucially, do not suffer from the sign problem. So the only obstacle to fully exploring these models is designing efficient updates for QMC simulations.

The author does a thorough job in exploring several different kinds of updates to the QMC operator string (the configuration), and demonstrates their efficiency for a wide range of parameters. A relevant application for this approach is presented at the end, where it is determined by measurements of the specific heat that there is an absence of a quantum spin liquid at the low temperatures that can be simulated.

I believe this work is solid, and deserves publication in SciPost Physics. However, I have some critiques as well. The biggest issue for me is that the majority of the paper is dedicated to technical details associated with the various updating schemes. I realize that the nature of this work is technical, and so this is unavoidable. However, for me, it significantly detracts from the readability of the paper. One solution to this issue would be to put technical details in appendices (as of now the paper has no appendices), and put executive summaries of each type of update in the main paper.

Another issue for me is that the efficiency of each update is only gauged by the acceptance probability, instead of the impact on autocorrelation times. The autocorrelation times are presented, but this is for all updates taken together (including parallel tempering) for different system sizes. It isn't clear which updates are the most important. It is possible the parallel tempering combined with one of the more simple updates is already sufficient to obtain the results presented in the paper.

Finally, the results occupy a rather small section at the end of the paper. The absence of the spin liquid phase can only be confirmed above a certain temperature (although it is a very low temperature). The paper would be improved if these results could be expanded on, perhaps by studying different observables beyond the specific heat.

Requested changes

I would say that the requested changes that I will propose are not entirely necessary, so it is up to the author how much they wish to address them. That being said, I think the paper could be substantially improved with the following modifications:

1) A simplified description of each type of update, where technical details of the implementation are moved to appendices.

2) The impact on autocorrelation times when including/excluding the various updates from the simulations.

3) A potential "minimal algorithm" that would be easiest for someone to implement and gives a reasonably good efficiency. This could include parallel tempering combined with one of the more simple updating schemes.

4) Some sort of computations beyond the specific heat that point to the absence of a spin liquid phase. Perhaps this is difficult, since traditional order parameters cannot identify this phase.

---

## Round 4 · Author Response

I have taken into consideration all the comments made by the referees and revised the manuscript based on their recommendations. Thanks to these changes, the manuscript is now more accessible to relevant readers. New numerical results are presented as additional evidence for the phase diagram discussed in Sec.4 and detailed pseudo-codes are provided in the appendices for readers who would like to write their own code. Additionally, detailed benchmarks for the efficiency of individual updates is included to clarify their capabilities. In conclusion, in my opinion this makes the manuscript suitable for publication in this journal.
Thank you,
Pranay Patil

---

## Round 4 · List of Changes

-
For each of Sec. 3.2, 3.3, 3.4, the details of the algorithm have been moved to the new appendices A,B,C (respectively), and discussion of autocorrelation functions is included for each update specifically along with data from simulations to support the claims in the discussions.
-
Pseudo-code for each update is now provided in the appendices. It is expected that following the pseudo-code will allow interested readers to easily develop their own version of the algorithm.
-
Data for entropy, dimer density and string order parameters is now presented in Sec.4 . This allows for a more comprehensive understanding of the phase diagram.
-
Minor typos and grammatical errors have been fixed after a re-reading.

---

## Round 5 · Referee Report · Anonymous (Referee 1) · 2025-3-21

Report

The author has responded appropriately to my last comments and to me, the manuscript appears ready for publication in this form.

Recommendation

Publish (easily meets expectations and criteria for this Journal; among top 50%)

---

## Round 5 · Author Response

Data for all plots available on Zenodo at https://zenodo.org/records/14922067

---

## Round 5 · List of Changes

1. Ordering of Fig 1 and 2 switched.
  2. Fig.6(a) moved to Appendix.
  3. All instances of Monte Carlo sizes have been explicitly labelled as "L=..." .
  4. The following sentences have been added after the first paragraph of Sec.4 : "In the phase diagram discussed below, we thus expect to find four phases. At high temperature, we expect a simple paramagnetic phase with maximal entropy and no signature in the order parameters we use to distinguish the other phases. At low temperature and $\Omega\gg\delta$, we expect the quantum paramagnet, which differs from the the high temperature one only in terms of entropy as it should have zero entropy for temperatures below the energy gap. At $\Omega\ll\delta$, we expect the classical spin liquid for $\Omega<T<\delta$, which has a non-zero entropy which is lesser than the maximal possible. In addition to the entropy, we also use a string order parameter to distinguish between the classical spin liquid and the quantum paramagnet as the former shows a non-zero value for strings of significant size whereas the latter does not. The quantum spin liquid is expected to emerge when we lower the temperature to $T\ll\Omega$ starting from the classical spin liquid regime, and this is characterized by zero entropy and a string order parameter behavior similar to that of the classical spin liquid. We do not find evidence of this last phase in the phase diagram which we are able to generate using the QMC simulations."

---

## Editorial Decision

accepted_in_target_journal